# Developing Guidelines for Thermal Comfort and Energy Saving during Hot Season of Multipurpose Senior Centers in Thailand

**Chorpech Panraluk * and Atch Sreshthaputra**

Department of Architecture, Chulalongkorn University, Bangkok 10330, Thailand; atch.s@chula.ac.th
* Correspondence: chorpech.p@student.chula.ac.th; Tel.: +66-816-749-069

**Abstract:** In Thailand, many government buildings and facilities are adapted to serve as Multipurpose Senior Centers (MSCs). However, most of them have been used without taking into account of thermal comfort of occupants. The present research aimed to develop guidelines for improving suitable indoor environment for the Thai elderly in hot season and analyze energy use of the 3 case-study MSCs. Both field study and climate-controlled chamber study were conducted. The obtained data were analyzed to develop the equation for predicting the thermal sensation, which would be inputted in the scSTREAM program for analysis purposes. The energy use was evaluated using the DOE-2 program. The results suggested that during 8:00 a.m.–12:00 p.m., natural ventilation should be used together with orbit fans to produce an actual air velocity of 0.64–0.73 m/s. From 12:00 p.m. to 4:00 p.m., air conditioners should set at 26.00–26.50 °C with an actual air velocity of 0.06–0.22 m/s. The results also showed that the developed guidelines could improve the level of thermal comfort from "slightly cool" to "neutral" and reduce energy use in hot season by 16.56% due to the reduction of cooling load and fan operation of air conditioning systems. Moreover, energy consumption in MSCs are also affected by the building parameters. These findings can be applied as guidelines for improving a large number of MSCs in Thailand.

**Keywords:** the Thai elderly; thermal comfort; energy saving; multipurpose senior center; computer simulation; hot season; sustainable development

## 1. Introduction

Thailand will become a complete aged society in 2021 and a super-aged society in 2031 [1,2]. As a lot of Thai elderly people in urban areas have to live alone without a caregiver, the concept of elderly day care service has emerged and widely presented [3]. The Thai government has encouraged the establishment of Multipurpose Senior Centers (MSCs) with the aim to provide elderly people with a place to do activities together during the daytime. Many of the government's buildings in Thailand (i.e., office buildings, community health centers, and public health centers) and facilities (e.g., multipurpose room, hall, and meeting room) have been adapted to be MSCs in order to support Thai elderly people. In Thailand, sustainable development, preparation for an aging society, and energy saving are among the major concerns reflected in the 12th National Economic and Social Development Plan [4]. Thailand has continuously supported the physical environment design that accommodates the needs of elderly people, according to the Ministerial Regulations on Specification of the Facility of Building for Disable or Deformed person and Aged Person B.E. 2548 [5]. However, related parties rarely pay attention to Indoor Environmental Quality (IEQ), which IEQ is important for the well-being of occupants [6]. Particularly, there are still few studies on thermal comfort of elderly people, which basically affects energy use in buildings and is also a key factor that can affect comfort, health,

and performance of elderly people [7]. At present, there are at least 879 MSCs [8] that are being used without taking into account of occupants' thermal comfort.

In countries with hot humid climate such as Thailand, the weather in hot season is sometimes intolerable. In order to reduce building heat, air conditioners are normally installed to control thermal indoor environment in buildings. The excessive use of air conditioners leads to an increase in electricity consumption [9–11]. Based on a statistical report, Thailand's electricity consumption is at its peak in hot season [12]. If air conditioners are not properly used, it will result in a waste of energy and uncomfortable outcomes. In Thailand, there has been a campaign to encourage people of all ages to set a set-point temperature of 25.0 °C. However, it may not be the ideal temperature for elderly people. Many previous studies indicated that elderly people have to confront physical changes such as a decline in the main sensory modalities [13], reduction in the density of sensory epidermal nerve fibers [14,15], change in the peripheral nerve system [16], and lower brown adipose tissue (BAT) [17,18], which will lead to changes in thermogenesis and thermal perception. In other words, elderly people tend to have different thermal sensations from other groups of people. This is consistent with Hwang and Chen [19], who conducted a study on thermal comfort of people of various ages and found that elderly people are likely to prefer warmer temperatures than younger adults. Schellen et al. [20], and Hoof and Hensen [21] reported that the elderly prefer higher temperatures than young adults by 0.5 °C and +2.0 K, respectively. According to Aguiar et al. [22], older individuals spend approximately 19–20 h/day indoors. Thus, it is essential to promote energy saving and enhance indoor thermal comfort for elderly people. In terms of energy saving, the use of natural ventilation is one vital strategy [23,24]. Kwong et al. [25] estimated that if the thermostat set-point is set 2.0 °C higher, a reduction of energy consumption can be achieved. In Thailand, although the government has determined a policy and prepared buildings to serve as MSCs, there is still little research on indoor environment adjustment for the Thai elderly. The objectives of this research were (1) to study the indoor thermal environmental conditions that the Thai elderly are satisfied with, (2) to develop the guidelines for improving thermal comfort in multipurpose senior centers, and (3) to analyze the energy use and comfort level resulting from the implementation of the proposed guidelines.

Regarding the evaluation of indoor environment, an experimental survey method is usually used to collect data about thermal comfort. The experimental survey method can be divided into 2 main approaches: the climate-controlled chamber study and the field study. Fanger [26] conducted the climate-controlled chamber study on the thermal comfort of people by adjusting human factors (i.e., clothing insulation ($I_{cl}$); and metabolic rate ($M_{et}$)), and environment parameters (i.e., air temperature ($T_a$); mean radiant temperature (MRT); relative humidity (RH); and air velocity ($V_a$)). The research proposed the relationship between the variables in order to assess the thermal comfort of people before developing into the Predicted Mean Vote (PMV) equation for general people. In terms of the field study, Nicol et al. [27] and other researchers [19,28–31] used the data obtained from field surveys to analyze the thermal comfort of people. This method was considered a basis for the development of Adaptive Comfort Standard (ACS) [32] that uses linear regression to analyze the relationship between indoor and outdoor environments. In addition, the evaluation of indoor environment and thermal comfort can be conducted using computational fluid dynamics (CFD) analysis. Cheong et al. [33] applied CFD software to evaluate the thermal comfort conditions of an air-conditioned theatre and measure the environmental parameters, including air temperature, air velocity, and relative humidity. Samiuddin and Budaiwi [34] also used CFD software to assess the thermal comfort in mosques in Saudi Arabia as it helped reduce costs and save time.

Based on this research gap, the present research aimed to explore the guidelines for improving thermal comfort in the MSCs that were adapted from the government's buildings. The results of this research will contribute to efficient energy use and can make elderly people stay in MSCs in a more comfortable way. This research used 3 methods: field study, climate-controlled chamber study, and computer simulation. The field survey was applied to evaluate the thermal comfort of the Thai elderly in case-study MSCs. However, in practice, it is impractical to adjust the thermal

environment in case-study MSCs in order to find an appropriate combination of environmental factors. Therefore, the experiments were carried out in a climate-controlled chamber rather than inside of a case-study MSC in use. The data obtained from the field study and the climate-controlled chamber study were analyzed to find the thermal environmental conditions that are suitable for the Thai elderly. The guidelines for improving indoor environment and thermal comfort were developed using CFD program (i.e., scSTREAM (Software Cradle, Tokyo, Japan)). The evaluation of energy use was carried out with the DOE-2 program (Lawrence Berkeley National Laboratory, Berkeley, USA). The finding of this research can be proposed as the guidelines to develop indoor thermal environmental conditions that the Thai elderly were preferable. Meanwhile, guidelines can be also helped to reduce energy use in buildings.

## 2. Materials and Methods

This research was conducted in hot season. The data collection took 3 months, starting from March to May 2018, in Phitsanulok City, whose climate is classified as Tropical Savanna (Aw) [35]. Phitsanulok City has received support from the Thai government and Japan International Cooperation Agency (JICA) to develop into a pilot zone for a sustainable ageing society [36]. The surveys were carried out in both the target case-study MSCs and climate-controlled chamber in order to find the range of indoor thermal environmental conditions that the Thai elderly feel comfortable. The statistical analysis and computer simulation techniques were used to analyze the obtained data.

### 2.1. Case Studies

The 3 case-study MSCs located in Phitsanulok City Municipality (Figure 1) that is divided by a river into two main parts: eastern and western. The case-study MSC 1 and 2 were located on the east side of the river, whereas the case-study MSC 3 was on the west side of the river. All case-study MSCs are parts of the public health centers.

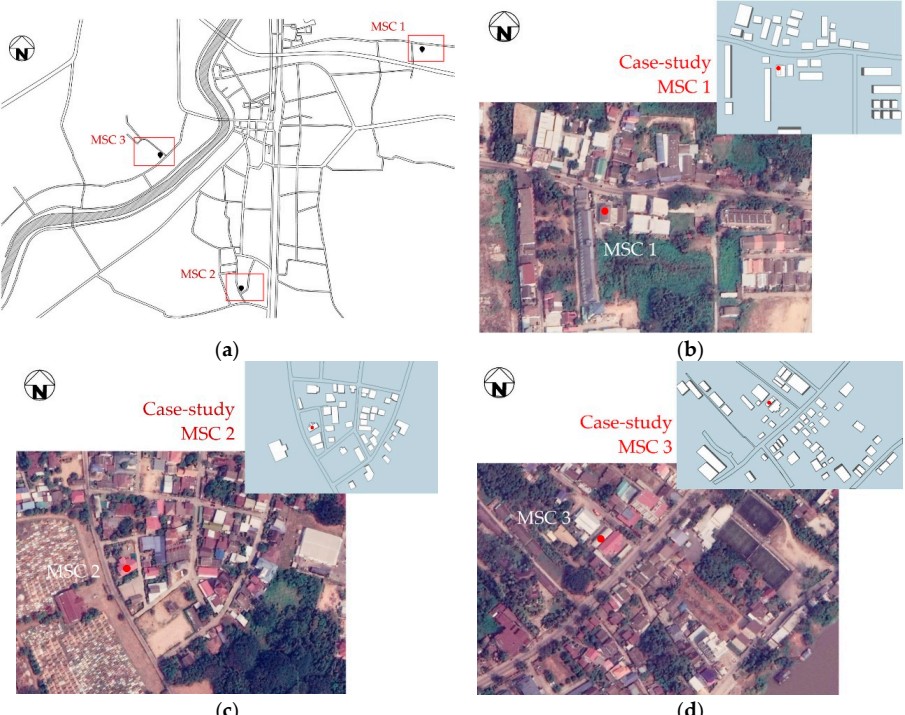

**Figure 1.** Case-study MSCs: (**a**) Their locations in Phitsanulok City; (**b**) surroundings of MSC 1; (**c**) surroundings of MSC 2; (**d**) surroundings of MSC 3.

### 2.1.1. Case-Study MSC 1

The case-study MSC 1 (Figure 2) was adapted from the community health center's meeting room, of which it is part of the public health center. It is located on the second floor. The short sides face the north-south axis. It has a wall and windows facing along the west that is the hottest direction. Dimensions of MSC 1 are 7.00 m (width) × 14.00 m (length) × 2.80 m (height).

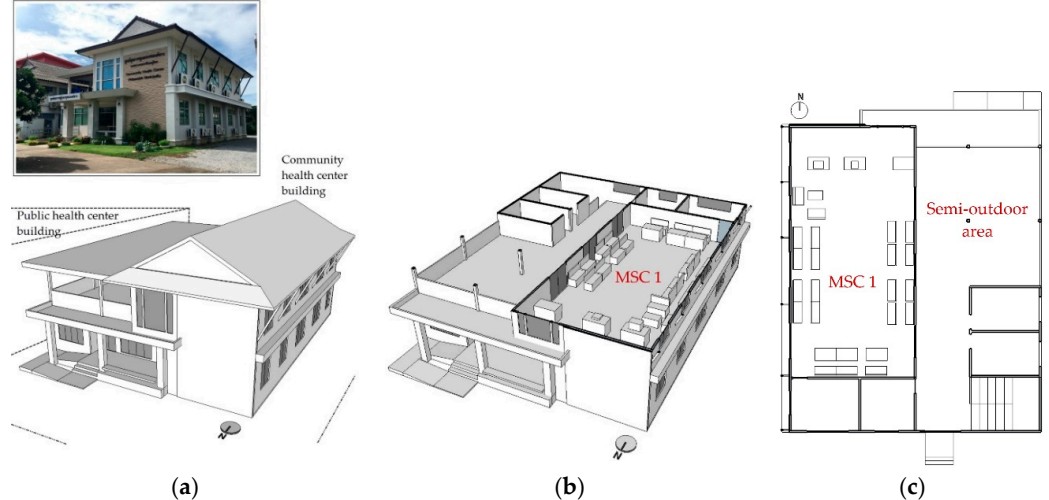

|(**a**)|(**b**)|(**c**)|

**Figure 2.** Case-study MSC 1: (**a**) The community health center building, where MSC 1 is located; (**b**) meeting room on the second floor, which was adapted to be MSC 1; (**c**) floor plan of MSC 1 and semi-outdoor area.

### 2.1.2. Case-Study MSC 2

The case-study MSC 2 (Figure 3), which was adapted from the public health center's hall, is located on the first floor. The short sides face the northeast-southwest axis. It looks similar to the U-shaped room. MSC 2 has overall dimensions approximately 7.50 m (width) × 10.50 m (length). Ceiling height is divided into two parts: 4.50 m and 2.75 m above the floor.

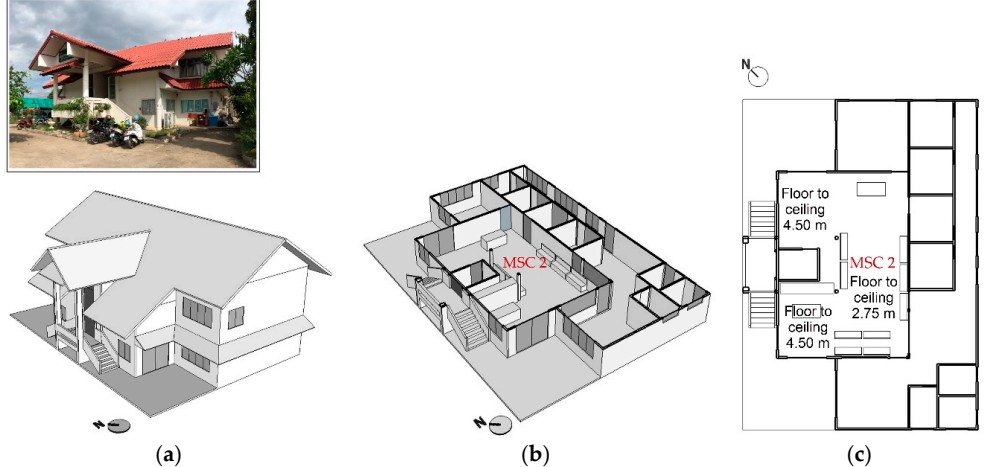

|(**a**)|(**b**)|(**c**)|

**Figure 3.** Case-study MSC 2: (**a**) The public health center building, where MSC 2 is Located; (**b**) hall on the first floor, which was adapted to be MSC 2; (**c**) floor plan of MSC 2.

### 2.1.3. Case-Study MSC 3

The case-study MSC 3 (Figure 4), which was adapted from the public health center's multipurpose room, is located close to the main building of the public health center. The short sides face the

northeast-southwest axis. It has dimensions of 6.15 m (width) × 14.50 m (length) × 2.80 m (height). Space is defined by a curved ceiling with the highest point of 3.80 m.

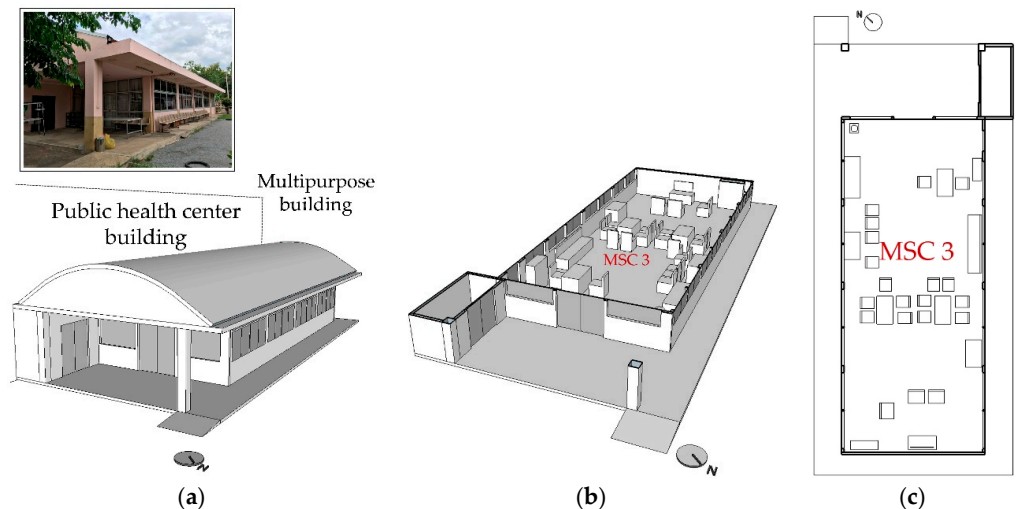

|       |       |       |
| :---: | :---: | :---: |
| (**a**) | (**b**) | (**c**) |

**Figure 4.** Case-study MSC 3: (**a**) Multipurpose building, where MSC 3 is located, was built separately nearby the public health center building; (**b**) multipurpose room, which was adapted to be MSC 3; (**c**) floor plan of MSC 3.

The MSC 1 looks similar to MSC 2 as they are rooms in the buildings, whereas the MSC 3 is a single building with around windows. However, the MSC 2 is located on the ground floor that is not received heat gain from the roof.

*2.2. Instruments, Data Collection Tools, and Software*

2.2.1. Instruments

This research used Tenmar 4002 to measure air temperature, relative humidity, and air velocity. Testo 445 and black globe thermometer were also applied to measure the mean radiant temperature. When collecting data in areas that were applied to the elderly do light activities (i.e., sitting, and both sitting and standing), indoor environments were measured at an approximate height of 0.73 m, which it was averaged from 0.60 m for sitting, and 0.85 m for both sitting and standing (activities in 15 min of the elderly).

2.2.2. Data Collection Tools

Two types of data collection tools that were used in this research were:

- The questionnaire (in Supplementary), which was comprised of 4 parts as follows:

  (1)　Part 1 aimed to gather information about indoor and outdoor thermal environments, based on the Thai Meteorological Department [37].
  (2)　Part 2 intended to elicit demographic data of the subjects.
  (3)　Part 3 was about the clothing of the subjects.
  (4)　Part 4 aimed to investigate the subjects' satisfaction towards the indoor thermal environments.

The ASHRAE (American Society of Heating, Refrigerating and Air-Conditioning Engineers) thermal sensation scale [38] and the scale for assessing humidity and air velocity sensation of Wang et al. [39] were adapted to use in this study. The details are as follows:

The ASHRAE 7-point scale (−3 = cold, −2 = cool, −1 = slightly cool, 0 = neutral, 1 = slightly warm, 2 = warm, 3 = hot), was used to examine the Thermal Sensation Vote (TSV).

The humidity sensation 7-point scale (−3 = very dry, −2 = dry, −1 = slightly dry, 0 = neutral, 1 = slightly humid, 2 = humid, 3 = very humid), was used to examine the Humidity Sensation Vote (HSV).

The air velocity sensation 7-point scale (−3 = very low, −2 = low, −1 = slightly low, 0 = neutral, 1 = slightly high, 2 = high, 3 = very high), was applied to examine the Wind Speed Sensation Vote (WSV).

- The document for recording the information of buildings, where the case-study MSCs are located, was used to collect data by researchers.

### 2.2.3. Software

This research applied the scSTREAM and DOE-2 Programs to simulate the guidelines. The details can be described as follow:

- scSTREAM is a thermo-fluid analysis software with a structured mesh, which is comprised of many small cuboids. It is most useful for applications where tiny details [40]. this study used scSTREAM to evaluate air flow and analyze thermal comfort inside the case studies.
- DOE-2, developed by University of California and Hirsch [41], is software that can predict the energy use for all types of buildings. This study used DOE-2 to evaluate energy use by inputting the building materials, operating schedules, and conditioning systems in a program to simulate the energy use of case studies both before and after implementing the developed guidelines.

### 2.3. Research Procedures and Data Analysis

This study was conducted in accordance with the tenets of the Declaration of Helsinki, and the experimental protocol was approved by the Ethical Committee of Naresuan University, Phitsanulok, Thailand (COA No. 344/2015). The research procedures are shown below.

From the flowchart in Figure 5, the details can be described as follows:

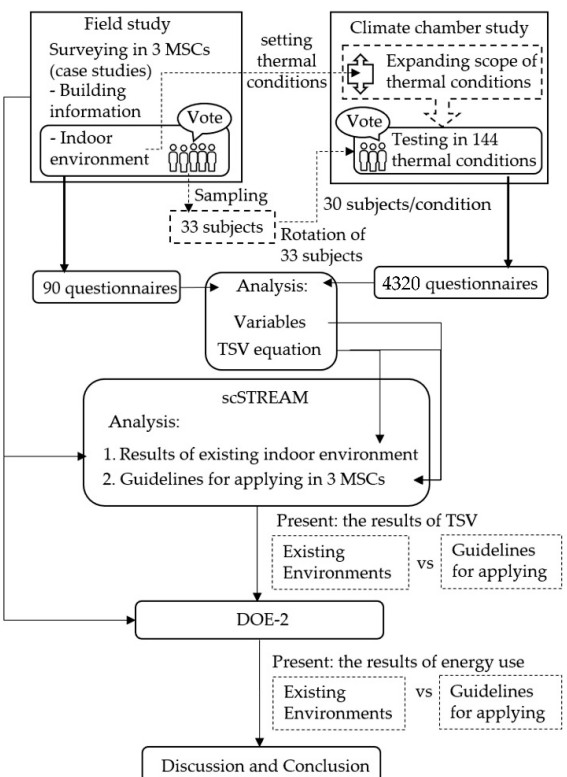

**Figure 5.** Flowchart of this research.

### 2.3.1. Field Study

As for the field survey, the demographic data, the sensation votes of subjects (i.e., TSV; HSV; and WSV), and indoor environments were collected using the questionnaires. The building information and thermostat set-point were also collected.

### 2.3.2. Climate-Controlled Chamber

In practice, it is not possible to adjust the thermal environment in the case-study MSCs in use in order to find suitable environment for the Thai elderly. Thus, the experiments in this study were carried out in a climate-controlled chamber instead of MSCs. This chamber was built in the education building of Naresuan University, which is located nearby the case studies. The chamber had dimensions of 3.00 m (width) × 4.65 m (length) × 2.35 m (height). The details are shown in Figure 6.

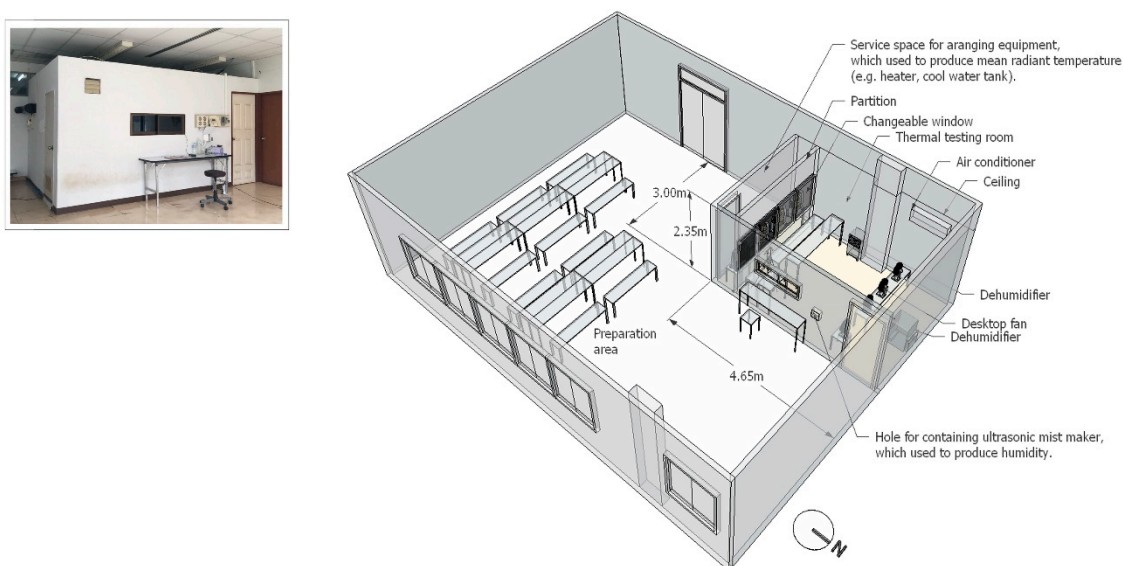

**Figure 6.** Climate-controlled chamber and details.

- Environmental condition setting in climate-controlled chamber

The indoor environment parameters obtained from the field study were used to further study the indoor environment parameters of the climate-controlled chamber. The subjects had to experience a total of 144 different thermal conditions within the chamber. The thermal conditions were determined from a combination of the following indoor environmental parameters.

(1) Parameter 1: Four levels of air temperatures produced by an air conditioner.
(2) Parameter 2: Four mean radiant temperatures values. When collecting data at the mean radiant temperature values that were higher than the room temperature, the sprayed gray and black aluminum windows would be installed together with 400 and 600 W infrared heaters. When collecting data at the lower mean radiant temperature values, the windows that contained ice and connected to the aluminum tube coils with cool water would be installed (Figure 7).

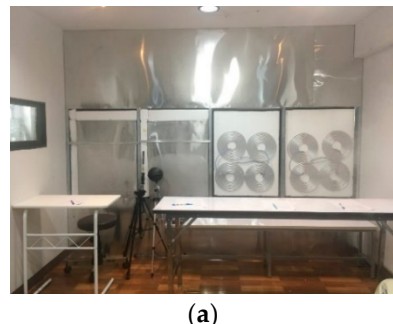 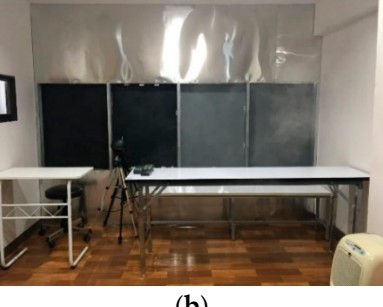

(**a**)　　　　　(**b**)

**Figure 7.** Climate-controlled chamber study: (**a**) The setting of mean radiant temperature that was lower than air temperature; (**b**) the setting of mean radiant temperature that was higher than air temperature.

(3)　Parameter 3: Three relative humidity levels. The 250 W ultrasonic mist maker was used to increase humidity, while 450 and 720 W dehumidifiers were used to decrease humidity.

(4)　Parameter 4: Three air velocity levels from 50 W fans.

The details of the thermal conditions are shown in Table 1.

**Table 1.** Mean ± standard deviation of conditioned levels in each parameter, which were used in the climate-controlled chamber in order to evaluate thermal sensation of the subjects.

| Level of Each Parameter | Air Temperature (°C) | Mean Radiant Temperature (°C) | Relative Humidity (%) | Air Velocity (m/s) |
|---|---|---|---|---|
| 1 | 21.55 ± 0.20 | $(T_a{}^1 − 2.46) ± 0.13$ | 45.28 ± 1.16 | 0.04 ± 0.02 |
| 2 | 24.02 ± 0.23 | $(T_a − 1.49) ± 0.16$ | 60.62 ± 1.03 | 0.51 ± 0.03 |
| 3 | 26.52 ± 0.17 | $(T_a + 1.40) ± 0.16$ | 74.75 ± 1.00 | 1.51 ± 0.03 |
| 4 | 28.95 ± 0.23 | $(T_a + 2.46) ± 0.15$ | | |

[1] $T_a$ is air temperature. Mean radiant temperature was adjusted to higher or lower than air temperature.

- The study in climate-controlled chamber

Thirty-three elderly people (16 males and 17 females) from the 3 case-study MSCs had to rotate and experience different thermal conditions in the climate-controlled chamber. Each thermal condition had to be experienced by 30 subjects. At least 28 subjects were required to experience all thermal conditions.

The chamber could accommodate 4 subjects at a time. During the experiment, the subjects were required to wear normal clothes, similar to what they wore in the case-study MSCs, and perform light activities with metabolic rates of 60–70 W/m$^2$ (or 1.0–1.2 met) for 15 min. If there was an increase in air velocity, whenever the subjects felt uncomfortable with the increased air velocity, they could decide to turn off the fans at any time. The response times were recorded.

2.3.3. Data Analysis

The 4230 questionnaires obtained from the climate-controlled chamber study and 90 questionnaires from the field survey were analyzed. The data analysis procedures can be described below:

- The data obtained from Part 1 to 3 of the questionnaires were analyzed to gain the results about the subjects' demographic information and outdoor thermal environments.
- The data obtained from Part 1 and 4 of the questionnaires, the indoor thermal environments that the subjects experienced in the case-study MSCs and climate-controlled chamber were analyzed and compared. The appropriate thermal conditions that can make the elderly people feel comfortable were evaluated by using average values of TSV; HSV; and WSV, which in this research, related values were identified as follows.

(1) Thermal Sensation Vote (TSV) is defined as an individual vote of occupant on a seven-point thermal sensation scale.

(2) Mean Thermal Sensation Vote (MTSV) is defined as the mean value of votes of a group of occupants on a seven-point thermal sensation scale.

(3) Humidity Sensation Vote (HSV) is defined as an individual vote of occupant on a seven-point humidity sensation scale.

(4) Mean Humidity Sensation Vote (MHSV) is defined as the mean value of votes of a group of occupants on a seven-point humidity sensation scale.

(5) Wind Speed Sensation Vote (WSV) is defined as an individual vote of occupant on a seven-point air velocity sensation scale.

(6) Mean Wind Speed Sensation Vote (MWSV) is defined as the mean value of votes of a group of occupants on a seven-point air velocity sensation scale.

(7) TSV and MTSV are used to assess the sensation related to temperature. HSV and MHSV are conducted to evaluate the sensation related to relative humidity while WSV and MWSV are employed to assess the sensation related to air velocity. The details of these sensation levels can be interpreted by mean values as follows.

The sensations can be described by using mean value range as in Table 2. Then, the statistical analysis method was used to determine the equation for predicting TSV in MSCs. The environmental variables in this research were relative humidity, air velocity, and operative temperature. The operative temperature ($T_o$) were calculated according to the ASHRAE standard-55 [38], and ASHRAE handbook [42].

**Table 2.** Mean value ranges that used to interpret sensations of MTSV, MHSV, and MWSV.

| Mean Value Range | MTSV | MHSV | MWSV |
|---|---|---|---|
| −3.00 to −2.50 | Cold | Very dry | Very low |
| −2.49 to −1.50 | Cool | dry | Low |
| −1.49 to −0.50 | Slightly cool | Slightly dry | Slightly low |
| −0.49 to +0.49 | Neutral | Neutral | Neutral |
| +0.50 to +1.49 | Slightly warm | Slightly humid | Slightly high |
| +1.50 to +2.49 | Warm | Humid | High |
| +2.50 to +3.00 | Hot | Very humid | Very high |

- The equation for predicting the TSV was inputted in the scSTREAM program to assess thermal comfort and evaluate appropriately indoor thermal environment for the Thai elderly. The feelings of the elderly in 3 case-study MSCs, before and after implementing the developed guidelines, were assessed and compared. The equations involved in the simulated program were similar to the equations used in the research of Huang et al. [43]. The details are as follows:

The kinetic energy equation of turbulence (k):

$$\frac{\partial(\rho k)}{\partial t} + \frac{\partial(\rho k u_i)}{\partial x_i} = \frac{\partial}{\partial x_j}\left[\left(\mu + \frac{\mu_i}{\sigma_k}\right)\frac{\partial k}{\partial x_j}\right] + G_k + G_b - \rho\varepsilon \tag{1}$$

The dissipation equation turbulence ($\varepsilon$):

$$\frac{\partial(\rho\varepsilon)}{\partial t} + \frac{\partial(\rho k\varepsilon)}{\partial x_i} = \frac{\partial}{\partial x_j}\left[\left(\mu + \frac{\mu_i}{\sigma_\varepsilon}\right)\frac{\partial\varepsilon}{\partial x_j}\right] + C_{1E}\frac{\varepsilon}{k} + (C_k + C_{3E}C_b) - C_{2E}\rho\frac{\varepsilon^2}{k} \tag{2}$$

where, $G_k$ and $G_b$ are the generation terms for kinetic energy k, caused by the average speed gradient and buoyancy, respectively; $C_{1E}$, $C_{2E}$, and $C_{3E}$ are empirical constants; $\sigma_k$ is the Prandtl number for k; and $\sigma_\varepsilon$ is the Prandtl number for $\varepsilon$.

- The DOE-2 program was used to evaluate the energy use in 3 case-study MSCs, before and after implementing the developed guidelines.

Finally, the appropriate set-point temperature for air conditioning, and actual air velocity both in air-conditioned space and in natural ventilated space were proposed to make the elderly feel comfortable while staying in the MSCs.

## 3. Results

The data analysis was carried out, after collecting data throughout the hot season. The analysis results can be described as follows.

### 3.1. General Information

The general information included the outdoor thermal environments and demographic data of the subjects. The details are shown below.

#### 3.1.1. Outdoor Environment Information

The outdoor environments (i.e., outdoor air temperature, outdoor relative humidity, and outdoor air velocity) were evaluated upon weather data from the Thai Meteorological Department [37] in the section of Phitsanulok Municipality, in which is study area. The descriptive statistics were used to analyzed the data.

Table 3 shows the outdoor environments during hot season. The average daily outdoor environments, comprising air temperature, relative humidity, and air velocity, were 29.48 °C, 69.71%, and 1.30 m/s respectively. The average outdoor air temperature, relative humidity, and air velocity would reach 29.56 °C, 69.30%, and 1.53 m/s respectively, starting from 8:00 a.m. to 12:00 p.m.

**Table 3.** The descriptive statistics of outdoor environments in hot season that used weather data of the Thai Meteorological Department.

| Outdoor Environment | Min [1] | Max [2] | M ± SD [3] |
|---|---|---|---|
| Daily outdoor air temperature (°C) | 21.00 | 38.60 | 29.48 ± 3.66 |
| Outdoor air temperature, 8:00 a.m.–12:00 p.m. (°C) | 22.80 | 34.70 | 29.56 ± 2.44 |
| Outdoor air temperature, 12:00 p.m.–4:00 p.m. (°C) | 25.20 | 38.60 | 32.97 ± 2.62 |
| Daily outdoor relative humidity (%) | 32.00 | 98.00 | 69.71 ± 14.85 |
| Outdoor relative humidity, 8:00 a.m.–12:00 p.m. (%) | 43.00 | 96.00 | 69.30 ± 10.71 |
| Outdoor relative humidity, 12:00 p.m.–4:00 p.m. (%) | 32.00 | 91.00 | 56.00 ± 11.26 |
| Daily outdoor air velocity (m/s) | 0.00 | 7.70 | 1.30 ± 1.25 |
| Outdoor air velocity, 8:00 a.m.–12:00 p.m. (m/s) | 0.00 | 4.60 | 1.53 ± 0.91 |
| Outdoor air velocity, 12:00 p.m.–4:00 p.m. (m/s) | 0.00 | 6.70 | 1.84 ± 1.00 |

[1] Min is minimum, [2] Max is maximum, and [3] M ± SD is mean ± standard deviation.

Then, the average outdoor air temperature and air velocity would rise to 32.97 °C and 1.84 m/s from 12:00 p.m. to 4:00 p.m., whereas the average outdoor relative humidity would decrease to 56.00%. Based on the obtained data, the natural ventilation should be used from 8:00 a.m. to 12:00 p.m. because the weather is not extreme.

#### 3.1.2. Demographic Information

Based on Table 4, the elderly people in all case-study MSCs had an average age, weight, height, and clothing insulation of 67.50–70.97 years, 59.17–60.67 kg, 155.60–156.80 cm, and 0.48–0.53 clo respectively. Similarly, the elderly people, who were the subjects of the experiments in the climate-controlled chamber, had an average age, weight, height, and clothing insulation of 65.85 years, 61.71 kg, 161.83 cm, and 0.51 clo respectively. Moreover, it was found that the elderly people in the case-study MSCs usually

perform light activities such as reading, singing, relaxing, and playing board games with an average metabolic rate of 66.67–67.50 W/m$^2$. Thus, those activities were also used during the experiments in the climate-controlled chamber. The results indicated that the subjects of the experiments similarly had an average metabolic rate of 67.21 W/m$^2$.

**Table 4.** The number of the subjects divided by gender, and mean ± standard deviation resulting from both field study and climate-controlled chamber study.

| Location | Male | Female | Age (year) | Weight (kg) | Height (cm) | Clothing Insulation Values (clo) | Metabolic Rate (W/m$^2$) |
|---|---|---|---|---|---|---|---|
| MSC 1 | 8 | 22 | 69.83 ± 8.09 | 60.67 ± 8.38 | 156.80 ± 7.37 | 0.51 ± 0.07 | 67.50 ± 3.46 |
| MSC 2 | 5 | 25 | 70.97 ± 8.50 | 59.17 ± 10.25 | 155.60 ± 5.62 | 0.48 ± 0.06 | 66.67 ± 3.43 |
| MSC 3 | 7 | 23 | 67.50 ± 7.86 | 59.88 ± 8.29 | 156.03 ± 6.86 | 0.53 ± 0.08 | 67.33 ± 3.46 |
| Chamber | 16 | 17 | 65.85 ± 6.85 | 61.71 ± 9.61 | 161.83 ± 8.27 | 0.51 ± 0.08 | 67.21 ± 3.51 |

Due to the increasing of age, most of the elderly confront body changing. Weight and height are the basis to calculate body mass index (BMI) and body surface area (BSA) using Du Bois method. BMI is a body size's measurement index. According to World Health Organization (WHO) classification's criteria for Asian-Pacific's population; BMI value between 23.0–24.9 kg/m$^2$ is considered as overweight person. This study calculated BMI of the subjects from the average value of weight and height in Table 4. The calculated average BMI shows that the BMI of elderly in MSC 1, MSC 2, MSC 3, and chamber are 24.67 kg/m$^2$, 24.44 kg/m$^2$, 24.60 kg/m$^2$, and 23.56 kg/m$^2$ respectively. Moreover, BSA of elderly in the study group were also evaluated. The BSA value in MSC 1, MSC 2, and MSC 3, which elderly women in each group are more than elderly men, are 1.61 m$^2$, 1.58 m$^2$, and 1.59 m$^2$ respectively. While the BSA value of the study group in the chamber, which the number of elderly women and elderly men are similar. The BSA value of this group is 1.66 m$^2$.

In accordance with heat dissipation depends on the body surface, whereas heat production is proportionate to the body mass. The results indicate that even though the BSA of the study group are in normal classification, but the BMI values are in overweight classification; which can be implied that the heat dissipation of this group is considered general but the heat production may change due to the body mass, that calculated based on BMI, are higher than standard value. According to proportionately affect heat exchange with environments and elderly people have to encounter internal body changes, suggesting a possibility that the Thai elderly may need special thermal comfort.

### 3.2. Analysis of Indoor Environment Affecting the Elderly's Thermal Comfort

The results obtained from the field study and chamber study were compared and analyzed by using MTSV, MHSV, and MWSV to find the appropriate indoor thermal environmental conditions for MSCs. The length of responsive periods from turning on to turning off the fans was also used.

From Figure 8, the elderly people participated in the field study and climate-controlled chamber study had similar feelings towards operative temperature and air velocity. The elderly people were found to feel comfortable with a wide range of relative humidity. The suitable indoor thermal environments for the Thai elderly, comprising operative temperature, relative humidity, and air velocity, were 27.40–29.60 °C, 47.00–70.00%, and 0.05–0.78 m/s respectively. Furthermore, the length of responsive periods from turning on to turning off the fans that were decided by the subjects was found to increase according to operative temperature. It can be implied that in order to improve thermal comfort in the MSCs, it is suitable to use orbit fans for serving the Thai elderly in various operative temperature.

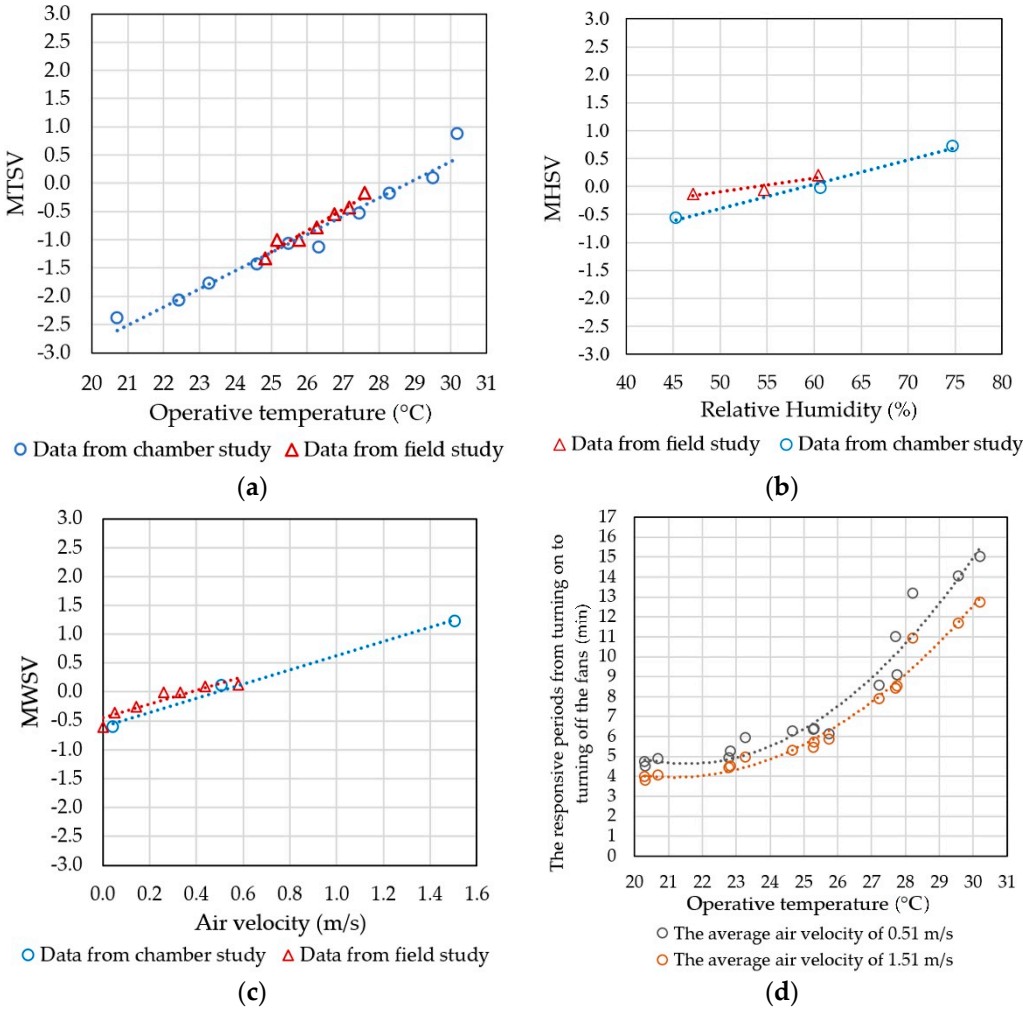

**Figure 8.** Comparison of feelings towards: (**a**) Operative temperature; (**b**) Relative humidity; (**c**) Air velocity. As for the experiments in climate-controlled chamber, fans were used to increase air velocity at 0.51 m/s and 1.51 m/s on various operative temperatures. (**d**) The relationship between operative temperature and responsive periods from turning on to turning off the fans that were decided by the subjects themselves.

The suitable indoor environments were simulated to determine the appropriate set-point temperature and actual air velocity in MSCs that make the elderly feel comfortable. However, many research studies indicated that elderly people require special thermal comfort that is different from other groups of people. Thus, the equation for predicting the TSV of the Thai elderly was further analyzed to evaluate the outcome of the indoor environment adjustment.

### 3.3. Analysis of TSV Equation for Predicting Thermal Sensation in Hot Season for MSCs

In this research, the elderly people participating in the climate-controlled chamber study were asked to wear normal clothes similar to what they wore when going to the case-study MSCs. Activities and all indoor thermal environmental conditions in case-study MSCs were also included in the climate-controlled chamber study. In order to determine an equation for predicting the TSV of the elderly people in MSCs, first of all, the multiple linear regression was conducted to analyze 90 data obtained from the field study as shown in Table 5.

**Table 5.** Analysis of data from the field study using multiple linear regression with stepwise.

| Variable | Model 1 | Model 2 |
|---|---|---|
| Constant | −9.088 *** | −9.122 *** |
| Operative temperature | 0.319 *** | 0.327 *** |
| Air velocity | | −1.057 *** |
| Relative humidity | | |
| $R^2$ | 0.260 | 0.397 |
| Adjusted $R^2$ | 0.251 | 0.383 |
| SE | 0.482 | 0.437 |

* $p < 0.05$; ** $p < 0.01$; *** $p < 0.001$, SE is the standard error for the unstandardized beta, and $R^2$ is coefficient of multiple determination, and Adjusted $R^2$ is a modified version of R-squared.

From Table 5, Multiple linear regression with stepwise were analyzed. It was found that there are 2 models for considering. SE values, which represents the average distance that the observed values fall from the regression line of model 1 (0.482) is higher than SE of model 2 (0.437). $R^2$ that is coefficient of multiple determination of model 2 (0.397) is higher than $R^2$ of model 1 (0.260). However, explanatory power of regression models should be considered, which is represented by Adjusted $R^2$ values. The value shows that Adjusted $R^2$ of model 2 (0.383) is higher than Adjusted $R^2$ of model 1 (0.251).

Thus, the best model, bases on some criteria (i.e., SE, $R^2$, Adjusted $R^2$), is presented by an equation for predicting thermal sensation vote from the field study ($TSV_f$) in hot season is as follows:

$$TSV_f = 0.327T_o − 1.057V_a − 9.122 \tag{3}$$

where, $TSV_f$ is thermal sensation vote from the field study, $T_o$ is operative temperature (°C), and $V_a$ is air velocity (m/s).

Analysis of the 4320 data obtained from climate-controlled chamber, the data obtained from the 30 subjects who tested in the similar indoor environment were averaged. The 144 data were analyzed using the multiple linear regression as shown in Table 6.

**Table 6.** Analysis of data from the climate-controlled chamber using multiple linear regression with stepwise.

| Variable | Model 1 | Model 2 | Model 3 |
|---|---|---|---|
| Constant | −8.889 *** | −8.537 *** | −8.877 *** |
| Operative temperature | 0.306 *** | 0.305 *** | 0.304 *** |
| Air velocity | | −0.459 *** | −0.457 *** |
| Relative humidity | | | 0.006 ** |
| $R^2$ | 0.825 | 0.907 | 0.913 |
| Adjusted $R^2$ | 0.824 | 0.906 | 0.911 |
| SE | 0.410 | 0.299 | 0.291 |

* $p < 0.05$; ** $p < 0.01$; *** $p < 0.001$, SE is the standard error for the unstandardized beta, and $R^2$ is coefficient of multiple determination, and Adjusted $R^2$ is a modified version of R-squared.

From Table 6, in terms of analysis an appropriate model, SE of model 1 (0.410) is higher than SE of model 2 (0.299) and SE of model 3 (0.291) respectively. The value of $R^2$ and Adjusted $R^2$ were evaluated and; have found that in model 1, $R^2$ (0.825) and Adjusted $R^2$ (0.824) is the lowest, while model 2, $R^2$ (0.907) and Adjusted $R^2$ (.906) are lower than model 3, which $R^2$ (0.913) and Adjusted $R^2$ (0.911) are the highest. Thus, model 3 was used.

The equation for predicting mean thermal sensation vote from the climate-controlled chamber study ($MTSV_c$) is as follows:

$$MTSVc = 0.304T_o - 0.457Va + 0.006RH - 8.877 \qquad (4)$$

where, $MTSV_c$ is mean thermal sensation vote from the climate-controlled chamber study, $T_o$ is operative temperature (°C), $V_a$ is air velocity (m/s), and RH is relative humidity (%).

Equations (3) and (4) were substituted by the values of existing environmental conditions in MSCs. The calculated data were compared using simple linear regression as shown in Figure 9.

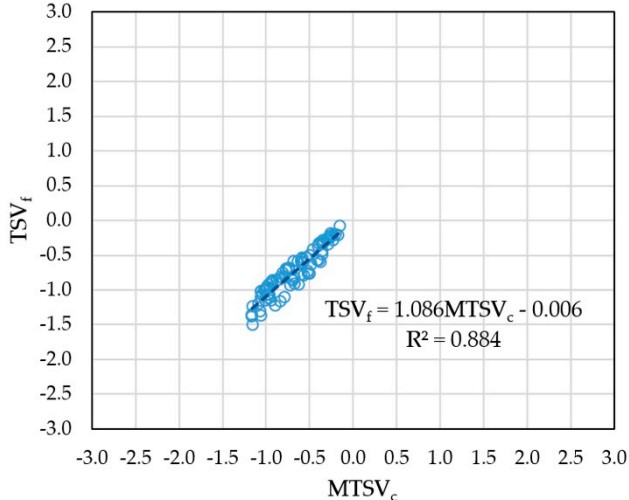

**Figure 9.** Comparison of $MTSV_c$ from climate-controlled chamber study and $TSV_f$ from field study.

From Figure 9, comparison of $MTSV_c$ and $TSV_f$, the equation for predicting the $TSV_f$ in the 3 case-study MSCs can be written as follows:

$$TSV_f = 1.086MTSV_c - 0.006 \qquad (5)$$

where, $TSV_f$ is thermal sensation vote from the field study, $MTSV_c$ is mean thermal sensation vote from the climate-controlled chamber study.

When replacing $MTSV_c$ with Equation (4), the equation can be written as follows:

$$TSV_f = 1.086 (0.304T_o - 0.457V_a + 0.006RH - 8.877) - 0.006 \qquad (6)$$

Therefore, the TSV prediction equation for using in all MSCs in hot season is as follows:

$$TSV = 0.330T_o - 0.496V_a + 0.007RH - 9.646 \qquad (7)$$

where, TSV is thermal sensation vote of the Thai elderly in multipurpose senior centers in hot season, $T_o$ is operative temperature (°C), $V_a$ is air velocity (m/s), and RH is relative humidity (%).

Equation (7) would be inputted in the scSTREAM to evaluate the thermal comfort of the Thai elderly, after improving the indoor environment according to the research results about the appropriate indoor thermal environment variables.

### 3.4. Analysis of Guidelines for Improving Thermal Comfort in the Case-Study MSCs Using the scSTREAM

3.4.1. Information of the Case-Study MSCs

In order to assess thermal comfort of occupants, a Computational Fluid Dynamics (CFD) program named scSTREAM [40] was used to perform simulations of air flow in and around building. In indoor

environments, the wind velocity was inputted in Equation (7) on the basis of operative temperature and relative humidity in the room. In this study, air velocity obtained from the measurement in MSC was inputted to compare with air velocity obtained from the analysis (in Figure 8) for considering and preparing appropriate environmental conditions in use. For building heat gain and energy consumption, an established public-domain simulation engine—DOE-2 [41]—was used to calculate the cooling load and thus energy use of the buildings. Operation schedules of air conditioners were adjusted in the program inputting based on the comfort assessment from the scSTREAM program.

As for this research, the information obtained from the environmental measurement together with collecting the actual votes of occupants in MSCs were shown in Table 7. The information on buildings was showed in Table 8.

**Table 7.** Air conditioner usage times, the range of thermal environmental conditions, and thermal sensations of the Thai elderly in the 3 case-study MSCs.

| Case Studies | Air Conditioner Usage Times | Operative Temperature (°C) | Relative Humidity (%) | Air Velocity (m/s) | MTSV | MHSV | MWSV |
|---|---|---|---|---|---|---|---|
| MSC 1 | 8:00 a.m.–4:00 p.m. | 25.10–27.50 | 48.30–63.80 | 0.00–0.60 | −0.70 | 0.00 | −0.33 |
| MSC 2 | 8:00 a.m.–4:00 p.m. | 24.82–26.25 | 44.30–52.10 | 0.00–0.48 | −0.97 | −0.13 | −0.27 |
| MSC 3 | 8:00 a.m.–4:00 p.m. | 26.10–27.75 | 52.40–55.20 | 0.00–0.62 | −0.53 | −0.03 | −0.20 |

**Table 8.** The building information of the 3 case-study MSCs.

| Topic | Details | MSC 1 | MSC 2 | MSC 3 |
|---|---|---|---|---|
| Shape parameter | Gross room area (m$^2$) | 98 | 72 | 89 |
| | Window area (m$^2$) | 12 | 9 | 33 |
| | Conditioned room area (m$^2$) | 98 | 72 | 89 |
| | Overall WWR [1] (%) | 9.7 | 4.8 | 27.1 |
| | North WWR (%) | 7.3 | 4.4 | 31.9 |
| | Non-North WWR (%) | 10.2 | 4.9 | 24.5 |
| Thermal parameter | U-Value of wall (W/m$^2$·°C) | 2.88 | 2.88 | 2.88 |
| | U-Value of window (W/m$^2$·°C) | 5.80 | 5.62 | 5.62 |
| | SHGC [2] of window | 0.55 | 0.50 | 0.46 |

[1] WWR is window-to-wall ratio, [2] SHGC is solar heat gain coefficient.

In Table 7, the result showed that the 3 case-study MSCs used air conditioners from 8:00 a.m. to 4:00 p.m., thermal conditions of all MSCs, comprising operative temperature, relative humidity, and air velocity, were 24.82–27.75 °C, 44.30–63.80%, and 0.00–0.62 m/s respectively. The thermal sensations of the elderly in all case-study MSCs had similar feelings. The actual MTSV, MHSV, and MWSV, were "slightly cool" (−0.97 to −0.53), "neutral" (−0.13 to 0.00), and "neutral" (−0.33 to −0.20) respectively.

From the information presents in Table 8 shows area that is used as conditioned room of MSC 1 (98 m$^2$) is more than MSC 3 (89 m$^2$) and MSC 2 (72 m$^2$) respectively; while window area of MSC 3 (33 m$^2$) is more than MSC 1 (12 m$^2$) and MSC 2 (9 m$^2$) respectively. Overall window-to-wall ratio (WWR) of MSC 3 (27.1%) is also more than MSC 1 (9.7%) and MSC 2 (4.8%). The north WWR, which is significant for lower energy use, shows that only the calculated ratio on the north WWR (31.9%) of MSC 3 is more than non-north WWR (24.5%), but in MSC 2 and MSC 1 show different value. The north WWR of MSC 2 (4.4%) and MSC 1 (7.3%) are less than non-north WWR, as the non-north WWR of MSC 2 is 4.9% and MSC 1 is 10.2%.

In addition, the results show that U-value of wall in all MSCs is the same value (2.88 W/m$^2$·°C), whereas, the U-value of windows of MSC 1 (5.80 W/m$^2$·°C) is higher than MSC 2 (5.62 W/m$^2$·°C) and MSC 3 (5.62 W/m$^2$·°C). As well as SHGC value, which is solar heat gain coefficient, in MSC 1 (0.55) also shows the highest value. While the value of SHGC of windows in MSC 2 (0.50) is higher than MSC 3 (0.46). However, the lowest value of both U-value and SHGC is better than higher value.

As for Table 8, the data about shape parameter and thermal parameter of case-study MSCs were conducted to input in the scSTREAM program in the next step.

### 3.4.2. Analysis of the Existing Natural Ventilation in the Case-Study MSCs

From 8.00 a.m. to 12:00 p.m., the average outdoor air temperature of 29.56 °C and outdoor relative humidity of 69.30%. When these values are substituted in the TSV equation that operative temperature is replaced by outdoor air temperature, it was found that the air velocity of 0.24–0.78 m/s can make the elderly feel "neutral". Then, natural ventilation flowing through the opened windows and doors of the 3 case-study MSCs were evaluated. Natural ventilation was simulated based on average outdoor wind speed at 1.53 m/s from the southwest. The results of air velocity and direction at 0.85 m above the floor that is in line with opened window heights are shown as follows.

According to Figure 10, in the case-study MSC 1, the air flew in with the velocity of 0.03–0.15 m/s. Air velocity of MSC 1 shows the lowest values from all MSCs. In the case-study MSC 2 (Figure 11), the air flew in the room and thoroughly reached all of the elderly with the velocity of 0.05–0.25 m/s. However, the air flowing in did not reach the corners of the room. In the case-study MSC 3 (Figure 12), the air flowing in the room was influenced by the adjacent building. When the windows near the adjacent building were open, the air would flow to the area around the northwestern wall with the velocity of 0.04–0.21 m/s, whereas the air velocity in the middle of the room was only at 0.04 m/s. It was found that some air velocity measured was still below 0.24 m/s, which made the elderly feel uncomfortable. Thus, the case-study MSCs should be used natural ventilation together with fans to produce adequate air velocity during 8:00 a.m.–12:00 p.m.

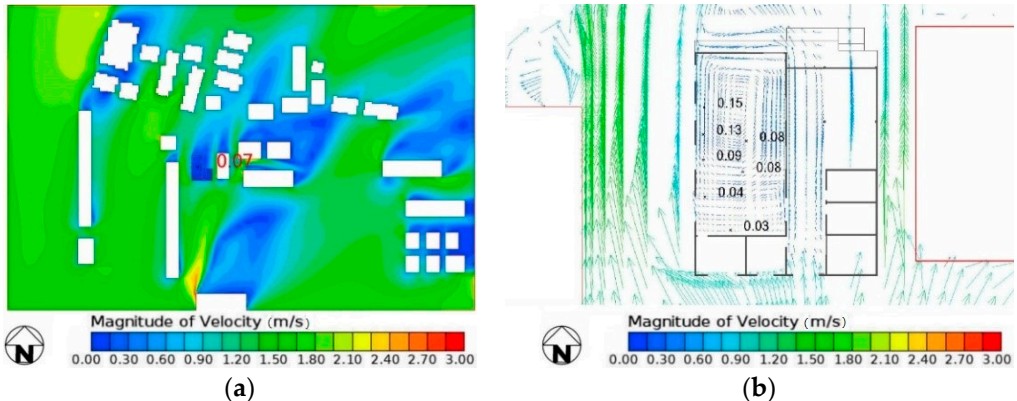

**Figure 10.** Natural ventilation of the case-study MSC 1: (**a**) Ambient air flow nearby the building, where MSC 1 is located; (**b**) air direction and air velocity of 0.03–0.15 m/s in MSC 1 space.

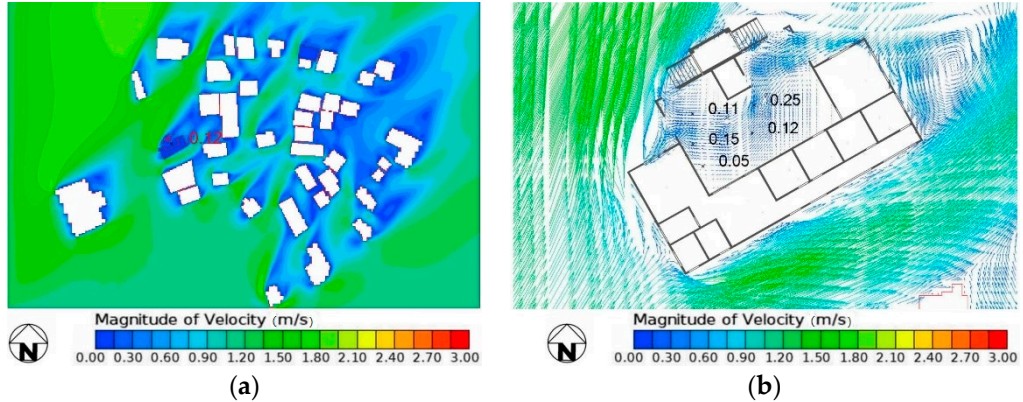

**Figure 11.** Natural ventilation of the case-study MSC 2: (**a**) Ambient air flow nearby the building, where MSC 2 is located; (**b**) air direction and air velocity of 0.05–0.25 m/s in MSC 2 space.

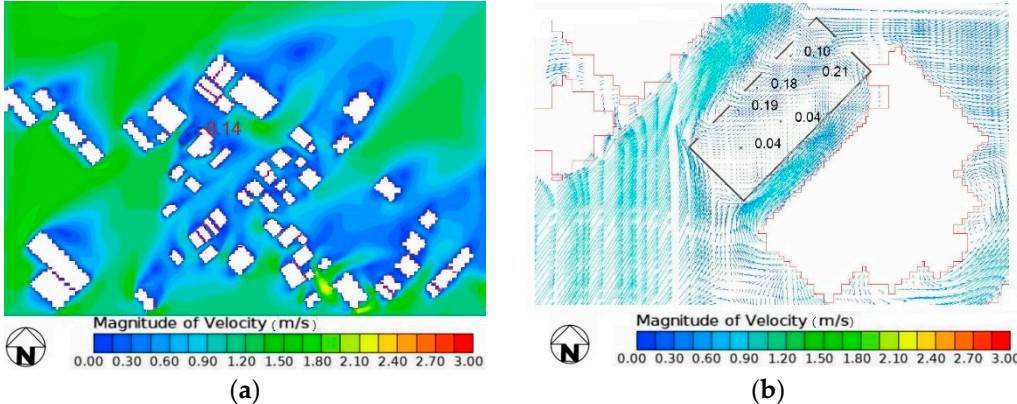

**Figure 12.** Natural ventilation of the case-study MSC 3: (**a**) Ambient air flow nearby the building, where MSC 3 is located; (**b**) air direction and air velocity of 0.04–0.21 m/s in MSC 3 space.

### 3.4.3. Analysis of Guidelines for Improving Indoor Thermal Comfort

The obtained TSV equation for hot season was applied to evaluate the thermal sensation of the Thai elderly in case-study MSCs using scSTREAM program. The clothing insulation and metabolic rate were determined at 0.48–0.53 clo and 60–70 W/m$^2$ respectively. Results of TSV in existing environments and after adjusting indoor environments based on the developed guidelines were examined and compared in two time ranges: 8:00 a.m.–12:00 p.m. and 12:00 p.m.–4:00 p.m. The indoor environments were measured at a height of 0.73 m above the floor. The ranges of TSV were interpreted similarly to MTSV ranges in the previous section.

From Figures 13–18, after all related environment variables were simulated based on the results of the survey, the TSV prediction equation was used to examine the feelings of the elderly in the 3 case-study MSCs. The results showed that the elderly felt "slightly cool", which was consistent with the average of actual vote resulting from the field survey. In this research, when the simulation technique was used to simulate the actual indoor environments based on an appropriate range of environmental conditions, it was found that the elderly's TSV resulting from the scSTREAM software was "neutral". Thus, the adjustment guidelines were created as follows.

From 8:00 a.m. to 12:00 p.m., in order to make the elderly feel comfortable, the 3 case-study MSCs should use natural ventilation together with orbit fans to add the air speed and cause actual air velocity of 0.64–0.73 m/s. The case-study MSC 1 and MSC 2 should install 3 and 5 orbit ceiling fans respectively. The case-study MSC 3, 5 orbit wall fans should be installed for improving the air flow through the room.

From 12:00 p.m. to 4:00 p.m., all case-study MSCs should use air conditioners to create thermal comfort. The case-study MSC 1 and MSC 3 should set the set-point temperature at 26.00 °C to create the operative temperature of 27.94–28.59 °C and set the air speed from air conditioners to create the actual air velocity of 0.14–0.22 m/s. The case-study MSC 2 should set the set-point temperature at 26.50 °C to create the operative temperature of 27.78 °C and set the air speed causing actual air velocity of 0.06 m/s.

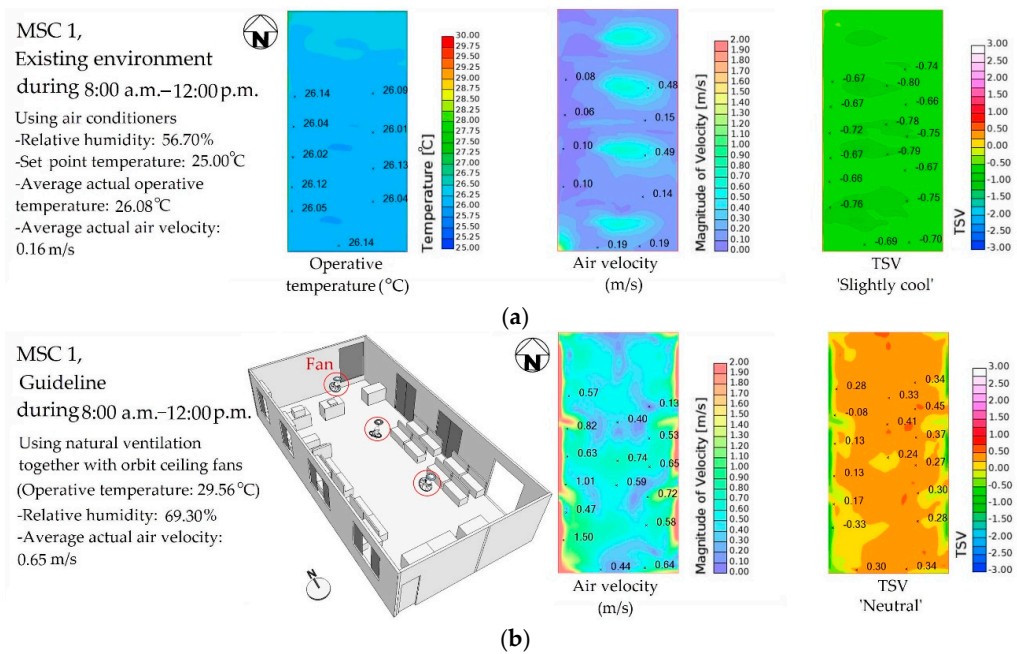

**Figure 13.** From 8:00 a.m.to 12:00 p.m., the simulated results of the case-study MSC 1 are as follows: (**a**) In the existing environment, air-conditioned space with actual operative temperature of 26.08 °C, which was caused by set-point temperature of 25.00 °C, and actual air speed of 0.20 m/s made the elderly feel "slightly cool"; (**b**) the guideline for using natural ventilation and orbit ceiling fans with actual air speed of 0.65 m/s made the elderly feel "neutral".

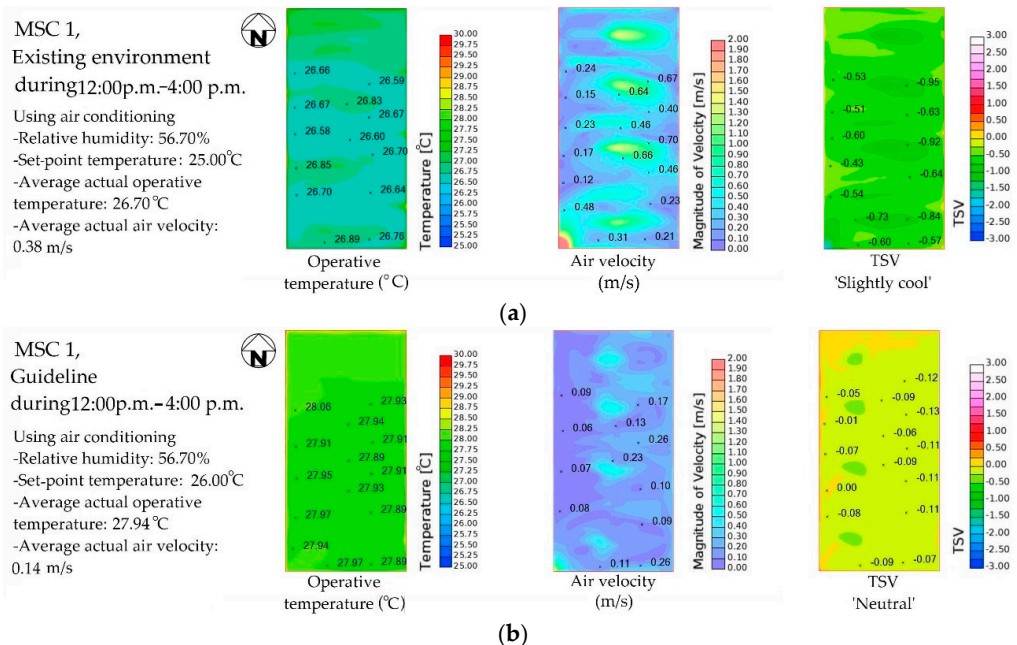

**Figure 14.** From 12:00 p.m.to 4:00 p.m., the simulated results of the case-study MSC 1 are as follows: (**a**) In the existing environment, air-conditioned space with actual operative temperature of 26.70 °C, which was caused by set-point temperature of 25.00 °C, and actual air velocity of 0.38 m/s made the elderly feel "slightly cool"; (**b**) the guideline for using air conditioning with actual operative temperature of 27.94 °C, which is caused by set-point temperature of 26.00 °C, and actual air velocity of 0.14 m/s made the elderly feel "neutral".

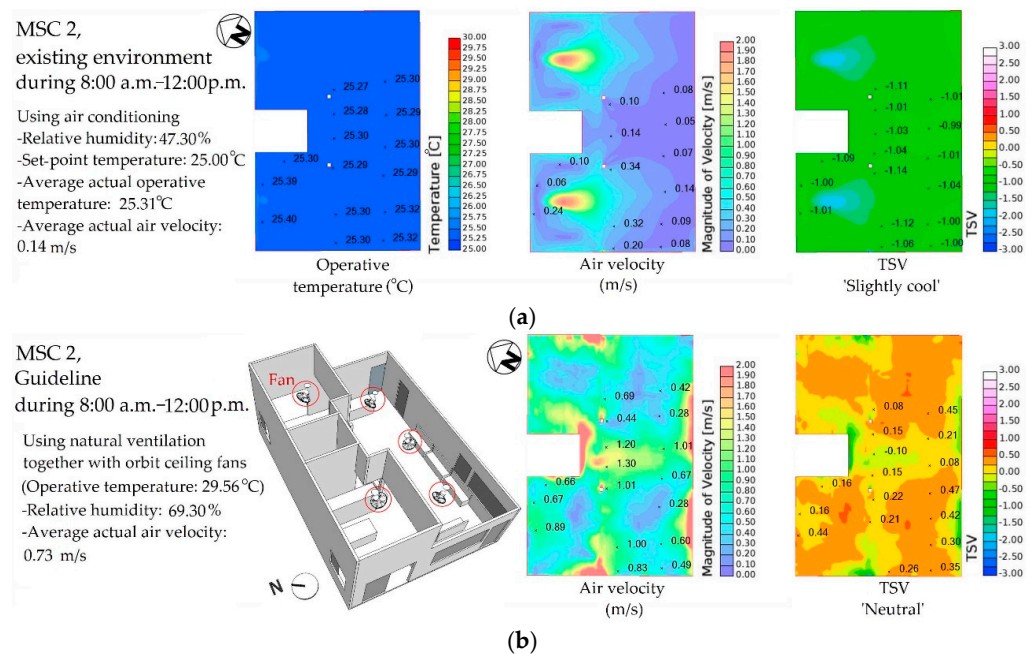

**Figure 15.** Results of the case-study MSC 2 from 8:00 a.m. to 12:00 p.m.: (**a**) The existing environment, air-conditioned space with actual operative temperature of 25.31 °C, which was caused by set-point temperature of 25.00 °C, and actual air velocity of 0.14 m/s made the elderly feel "slightly cool"; (**b**) the guideline for using natural ventilation and orbit ceiling fans with actual air velocity of 0.73 m/s caused "neutral" sensation.

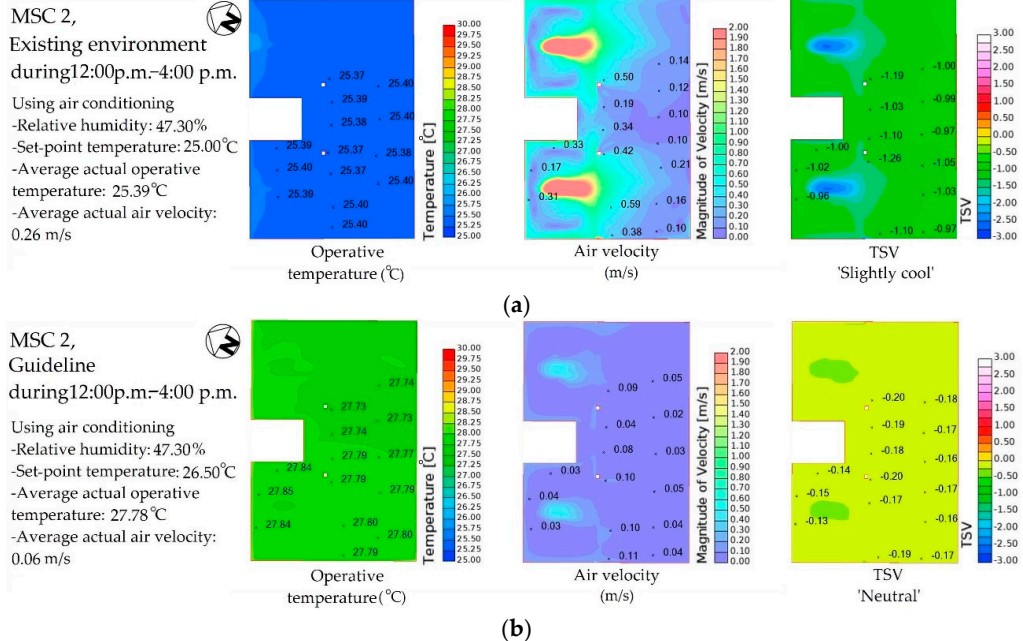

**Figure 16.** Results of the case-study MSC 2 from 12:00 p.m. to 4:00 p.m.: (**a**) The existing environment, air-conditioned space with actual operative temperature of 25.39 °C, which was caused by set-point temperature of 25.00 °C, and actual air velocity of 0.26 m/s caused "slightly cool" sensation; (**b**) the guideline for using air conditioning with actual operative temperature of 27.78 °C, which was caused by set-point temperature of 26.50 °C, and actual air velocity of 0.06 m/s caused "neutral" sensation.

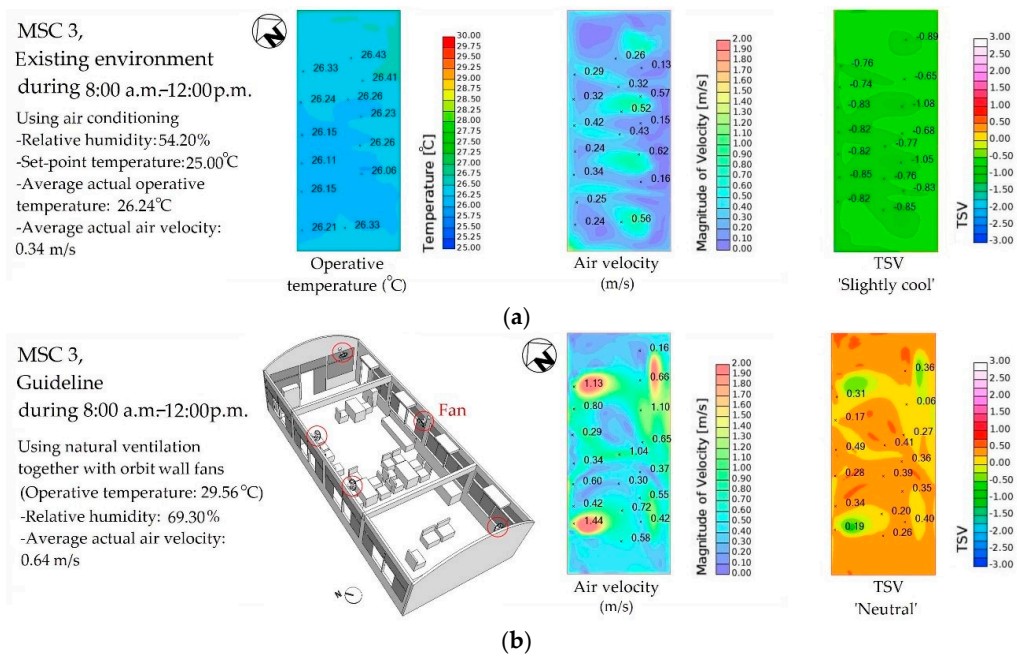

(a)

(b)

**Figure 17.** Results of the case-study MSC 3 from 8:00 a.m. to 12:00 p.m.: (**a**) The existing environment, air-conditioned space with actual operative temperature of 26.24 °C, which was caused by set-point temperature of 25.00 °C, and actual air velocity of 0.34 m/s caused "slightly cool" sensation; (**b**) the guideline for using natural ventilation and orbit wall fans with actual air velocity of 0.64 m/s caused "neutral" sensation.

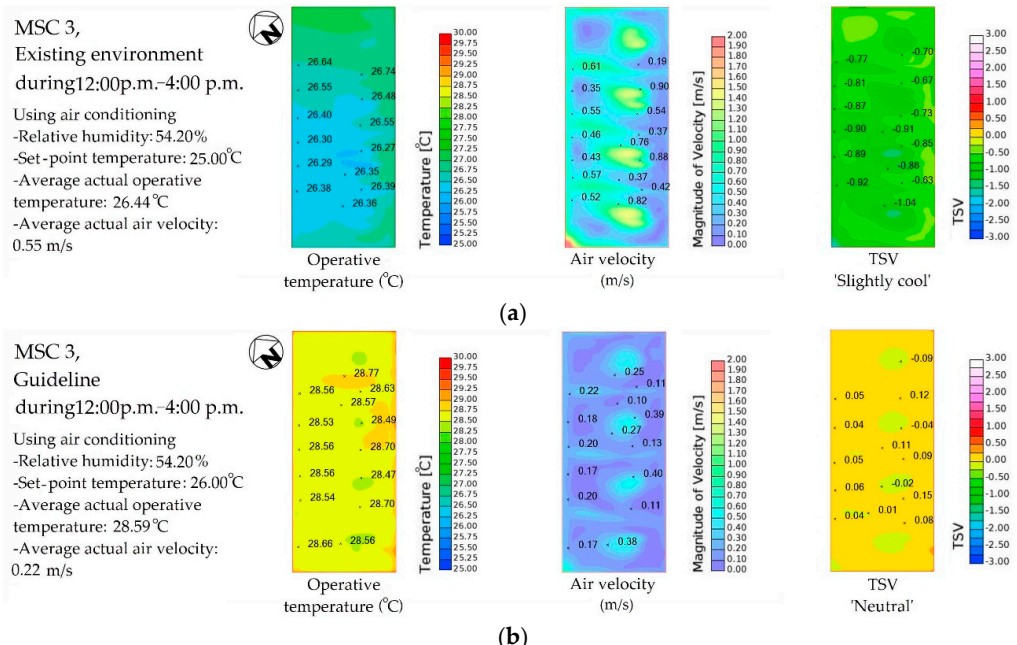

(a)

(b)

**Figure 18.** Results of the case-study MSC 3 from 12:00 p.m. to 4:00 p.m.: (**a**) The existing environment, air-conditioned space with actual operative temperature of 26.44 °C, which was caused by set-point temperature of 25.00 °C, and actual air velocity of 0.55 m/s made the elderly feel "slightly cool"; (**b**) the guideline for using air conditioning with actual operative temperature of 28.59 °C, which was caused by set-point at 26.00 °C, and actual air velocity at 0.22 m/s caused "neutral" sensation.

### 3.5. Evaluation of Energy Use

The guidelines for adjusting the indoor environment included using natural ventilation together with fans and using air conditioning with the appropriate setting. The energy use (i.e., monthly electricity, electric end uses) was evaluated using the DOE-2 program. The results are as follows.

Figure 19 shows the energy use of the 3 case-study MSCs in hot season (early March–end of May). After implementing the developed guidelines, Values of monthly electricity consumption in hot season were reported to decrease from 4031 kWh to 3490 kWh for MSC 1, from 3034 kWh to 2472 kWh for MSC 2, and from 3234 kWh to 2631 kWh for MSC 3.

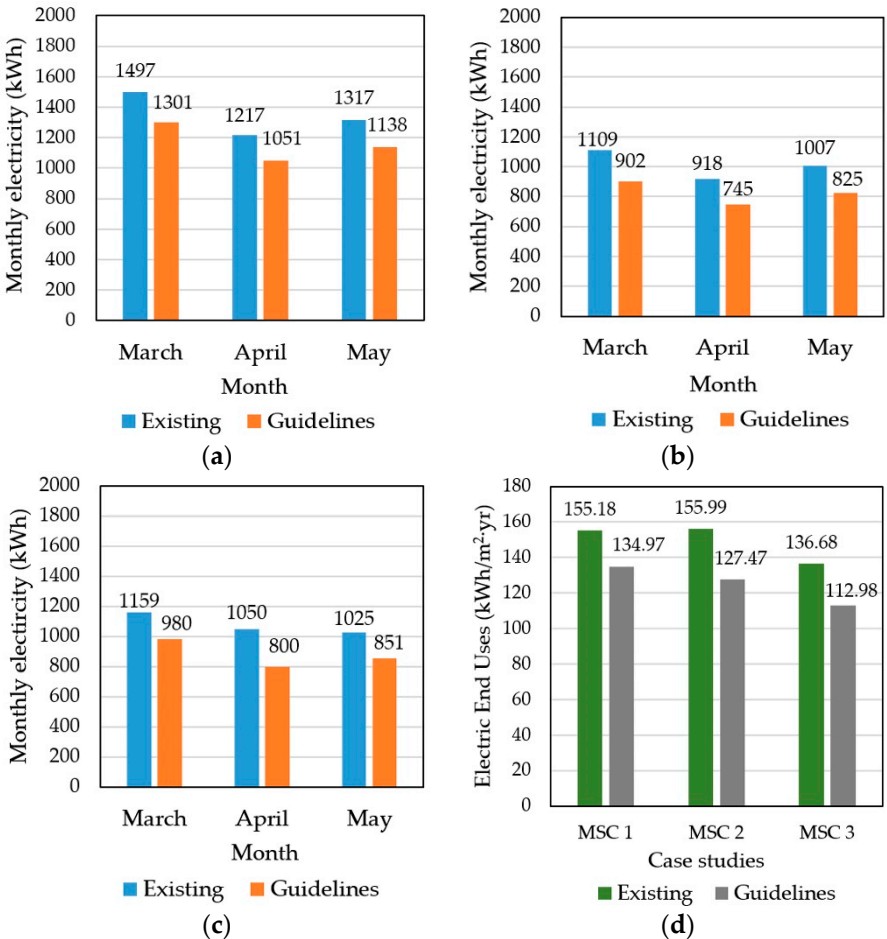

**Figure 19.** Comparison of the energy use before and after implementing the developed guidelines: (**a**) Monthly electricity consumption of MSC 1; (**b**) monthly electricity consumption of MSC 2; (**c**) Monthly electricity consumption of MSC 3; (**d**) electric end uses of the 3 case-study MSCs.

In addition, Values of electric end uses, which are results of electric use per square meter throughout the year, were also reported to decrease from 155.18 kWh/m$^2$·yr to 134.97 kWh/m$^2$·yr for MSC 1, from 155.99 kWh/m$^2$·yr to 127.47 kWh/m$^2$·yr for MSC 2, and 136.68 kWh/m$^2$·yr to 112.98 kWh/m$^2$·yr for MSC 3. The results also indicated that MSCs, in which MSC 3 is the most effective energy use, while MSC 2, and MSC 1 are less effective respectively.

Although MSC 3 has a lot of windows with glass that admitted solar radiation into building and converting it into heat gains, energy consumption in MSC 3 is still the most effective. With regard to orientation (i.e., the short sides face the northeast-southwest axis and usage of a solid wall in the southwest direction), and location (i.e., usage the advantage of nearby buildings for shielding solar radiation), these parameters are significant for energy use of MSC 3. As for building elements (i.e., long eaves, window ratio in the north, and appropriate window sizes for ventilating and wind-receiving),

and building materials (i.e., usage of window glass that U-value and SHGC are lower than MSC 1 and MSC 2), are important for resisting heat gains. Moreover, a lot of flexible windows that can adapt voids, are also an advantage to natural ventilation.

In terms of MSC 2, which its form is similar to U-shape, which some part has double space. Before implementing developed guidelines, MSC 2 used air conditioner all day long. The value of electric end uses of MSC 2 is similar to MSC 1, which value is the highest, because of the effect of double space in MSC 2. After implementing developed guidelines, natural ventilation is carried out to use in the morning time. As the orientation and location on the first floor of MSC 2 affects reducing heat gains, the electric end uses of MSC 2 is inferior to MSC 1, which is different from the electric end uses before implementing developed guidelines.

MSC 1 consumes the highest energy, As the orientation that the long sides face the east-west axis, heat gains from the west are a weakness to energy use. While the usage of natural ventilation, its orientation also affected air flows in MSCs that the result of air velocity shows the lowest values from all MSCs. In addition, the least effective window glass is also disadvantaged when compared with other MSCs.

Finally, from the simulation of DOE-2 program, the implementation of the developed guidelines could reduce energy use in hot season by 16.56% due to reduction of cooling load and fan operation of air conditioning systems.

## 4. Discussion

This research evaluated the energy use and thermal comfort of the Thai elderly in hot season using the comfort equation developed by field survey and laboratory test of occupants. The relevant variables, including operative temperature, relative humidity, and air velocity, and data of thermal sensation vote were also analyzed to develop thermal comfort spaces for the Thai elderly. Referring to the demographic information, although the BSA values are in normal classification as same as the average of general people overall [44], which was calculated in both men and women, the study group' s BMI values that are consistent with general Thai elderly in urban area are in overweight classification [45]. Thus, not only Thai elderly face internal body changes, but also body size changes, which may affect thermal sensation. It was found that the Thai elderly were satisfied with the operative temperature of 27.40–29.60 °C, air velocity of 0.05–0.78 m/s, and relative humidity of 47.00–70.00%. In this study, the CFD program (i.e., scSTREAM) was applied to analyze the relationship between the ambient indoor environment variables in the area where the elderly used and the setting of air conditioning, including set-point temperature and actual air velocity. In addition, those variables were also used to analyze the usage of natural ventilation and fans in MSCs.

The TSV equation in this research was inputted in the scSTREAM program to find the results of TSV in case-study MSCs. The results showed that the Thai elderly felt "slightly cool" when the existing indoor environments in computer simulation appeared as follows: operative temperature of 25.31–26.70 °C, relative humidity of 47.30–56.70%, and air velocity of 0.14–0.55 m/s. The results of TSV from computer simulation are consistent with the actual MTSV resulting from field survey, which might be because the TSV equation was already analyzed to be suitable for further use.

According to Cândido et al. [46] and Spentzou et al. [47] suggested that the use of natural ventilation is a strategy for saving energy and creating indoor thermal comfort. The results of this research indicated that during 8:00 a.m.–12:00 p.m., when the average outdoor air temperature is at 29.56 °C, average outdoor relative humidity is at 69.30%, and average outdoor air speed is at 1.53 m/s with the air flowing from the southwest, if all case-study MSCs use natural ventilation alone, the air velocity of 0.03–0.25 m/s within the room will not be able to make the elderly feel comfortable. Therefore, in order to make the elderly feel comfortable, the research results indicated that natural ventilation should be used together with orbit fans to produce an actual air velocity of 0.64–0.73 m/s.

From 12:00 p.m. to 4:00 p.m., when the outdoor air temperature is higher with an average of 32.97 °C, air conditioners should be used to provide thermal comfort. The set-point temperature should be 26.00–26.50 °C and the actual air speed in areas should be 0.06–0.22 m/s.

Moreover, the results from DOE-2 program showed that by adjusting the usage time of air conditioners and using natural ventilation together with orbit fans, the energy use in the 3 case-study MSCs in hot season was found to decrease by 16.56% due to reduced cooling load and fan operation of air conditioning systems, which is in line with the research of Park and Battaglia [48], and Malkawi et al. [49]. The usage of natural ventilation led to reduced energy consumption in buildings.

The results of this research can be used to directly benefit the 3 case studies and other MSCs with similar conditions in Thailand. Furthermore, the outcomes also demonstrate the cost-effectiveness of air conditioning usage behavior change, which are consistent with the research by Lim and Yun [50] on the influence of occupant factor. However, creating thermal comfort is associated with various factors, including the location of building [51], construction material, appropriate design such as better orientation and optimal window size [52]. New buildings should use simulation programs to carry out an analysis since the design stage [53]. Yildiz and Arsan [54] confirmed that building parameters affect energy consumption. The findings of previous studies are consistent with this study, which is considered from simulation results of DOE-2 program and building information. It is shown that in hot season, the highest energy consumption is MSC 1, while MSC 3 and MSC 2 consume less energy respectively. As for electric end uses, the highest values are MSC 1, MSC 2, and MSC 3 respectively. In addition, the energy consumption depended on the building parameters such as orientation, location, form and space, building elements, and building materials.

Orientation, the long sides of MSC 1 face the east-west axis. Wall and windows face along the west direction. It causes different heat gains in MSC 2 and MSC 3 that face the northeast-southwest axis. As for location, MSC 1 is located on the second floor. MSC 2 is located on the first floor. MSC 1 receives heat gains from the roof, whereas MSC 2 does not. MSC 3 is located on the ground floor near the public health center building. It uses the advantage of nearby building shade to protect solar radiation. Thus, although MSC 3 had a lot of windows, Electric end uses value is the lowest.

Regarding to architecture, form and space is important factors. MSC 2 looks similar to U-shape and has double space. If it uses air conditioning, it affects consuming more energy consumption. The results showed that before implementing developed guidelines, the electric end uses of MSC 2 is similar to MSC 1. However, after implementing developed guidelines, which the natural ventilation was adapt to be used, the energy use of MSC 2 is inferior to MSC 1, which shows the different result because of the effects of double space to cooling load. As for building elements (i.e., windows and eaves), window ratio and the window size in all MSCs are a significant factor for heat gains. North WWR affect lower energy use than non-north direction. Moreover, the flexible windows, revealed that they are an advantage to control natural ventilation. In terms of the eaves, the long eaves are able to reduce heat gains. Thus, the usage of elements in MSC 3 affects the electric end uses the least. As for building materials, the study revealed that window glass with effective U-value and SHGC is important for saving energy. The low effective window glass, which was installed in the west of MSC 1, causes consuming more electricity.

Thus, the study outcome can be summarized into 3 parts, which should be carried out to consider in order to develop MSCs in Thailand for its thermal comfort and energy saving as follows: technology, behavior change, and building design. Technology such as usage of fans and thermostat that the elderlies can use and observable respectively. While, behavior change such as usage of natural ventilation with orbit fans, instead of using air conditioning also should be conducted to use practically. However, orientation, location, form and space, building elements, and building materials should be considered as factors under building design and construction.

## 5. Conclusions

According to a number of older population in Thailand are growing up rapidly. The existing government buildings have been adapted to be MSCs in order to serve Thai elderly people. Nonetheless, MSCs have been applied without considering about thermal comfort of elderly occupants. This study was conducted to investigate indoor environmental conditions for Thai elderly, in order to develop guidelines for improving thermal comfort in MSCs, analyze the energy use as well as comfort level resulting from guidelines. In terms of strength of this research, this study employed 3 methods to study the propose objectives: Field study, climate-controlled chamber study and computer simulation. The contributions of this study are to find a comfort zone, which is a key factor that can affect health and performance of Thai elderly; and to find a potential design criterion which should be used for building improvements; for better result in thermal comfort and energy efficiency, as well as cost savings for Thai government.

The study found that the combination of environmental variables significantly influenced thermal comfort within the building. The elderly participants, who are overweight similar to general Thai elderly in urban area, are comfortable in special thermal environmental conditions as follows: operative temperature of 27.40–29.60 °C, relative humidity of 47.00–70.00%, and air velocity of 0.05–0.78 m/s. In unconditioned mode, using natural ventilation together with orbit fans with air speed at 0.64–0.73 m/s in the morning, thermal comfort is achievable. In air-conditioned mode in the afternoon, increasing the set-point temperature to 26.00–26.50 °C together with reducing air speed to 0.06–0.22 m/s. Thermal comfort is optimized. This can save cooling energy by 16.56%.

These findings suggest that following these operation guidelines could help reduce energy use in all MSCs yet providing better comfort. In terms of building design and construction; optimization of orientations, building forms and space planning, and building materials affected energy use in each MSC. Orientation affects heat gains inside MSCs. The position of MSC in the highest floor received heat gains from the roof which affect energy consumption. The cooling load will be increased in double height space. Furthermore, high-performance windows, exterior shading, and glass types window can help reduce radiative heat gains thus providing better comfort and energy saving.

As for the limitation, this study was conducted in only one climate region and investigated only in the light physical activities of the Thai elderly, which had metabolic rate at 60–70 W/m$^2$. Thus, it is recommended that future research should be performed in other climate regions of Thailand and examine in other physical activities of the Thai elderly.

**Supplementary Materials:** The following are available online at http://www.mdpi.com/2071-1050/12/1/170/s1. Developing Guidelines for Thermal Comfort and Energy Saving During Hot Season of Multipurpose Senior Centers in Thailand.

**Author Contributions:** Individual contributions from authors occurred as follows: conceptualization, C.P.; methodology, C.P.; formal analysis, C.P.; investigation, C.P.; writing—original draft preparation, C.P.; writing—review and editing, C.P.; supervision, A.S. All authors have read and agreed to the published version of the manuscript.

**Funding:** This research received no external funding.

**Acknowledgments:** The authors are grateful to Phitsanulok Municipality for coordinating the case-study MSCs and Naresuan University for supporting the building to locate the climate-controlled chamber. In addition, the authors would like to express sincere thanks to all the subjects for their invaluable collaboration, which contributes to the development of MSCs.

**Conflicts of Interest:** The authors declare no conflict of interest.

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
