# Peer review of "Developing Guidelines for Thermal Comfort and Energy Saving during Hot Season of Multipurpose Senior Centers in Thailand"

_sustainability, doi:10.3390/su12010170_

Round 1

Reviewer 1 Report

The paper is well-written and structured, and the aim of this research paper is suitable for publication in Sustainability journal. I just want to say that the paper structure has been designed exceptionally in order to demonstrate all findings from three case study buildings. It demonstrates rigorous amount of data findings and interpretations for highlighting significant importance of thermal comfort and its impact on energy use. I found that the case study buildings are interesting and undertaken methodology is novel in this particular field. I found that the scope of this paper is also suitable on the recent studies undertaken in this field by other scholars around the world. But the case study location demonstrates that hot and humid climate characteristics have had a significant impact on occupants’ thermal comfort in accordance with energy use. As a reviewer, I appreciated that all the findings are summarised in a paper format without distorting main objectives of the study.

Reviewer 2 Report

This article presents a developing guideline for thermal comfort and energy savings of multipurpose senior centers in Thailand. The work is very well organized and the article is in the scope of the journal. The methods used are very clear for readers. However, I will ask for a major revision, just to enhance and justify the following points:

Table 2 shows some important information that could be used by other readers, hence, the author needs to highlight the source of all information presented in this Table. This can be solved in Lines 266 to 268.

As the questionnaire is the main source of information for this work, I suggest adding it as a supplementary document where the reader could make an access and prove the discussion. 

In Table 3, what is the real effect of Age, Weight, and Height on the final results.

In line 294, the author mentioned the term Mean Thermal Sensation Vote (MTSV) for the first time in the article. Please make sure to define this term in the materials and methods section.

Lines 295 to 302, a confused paragraph. it would be more comfortable to organize this information in a proper Table or Figure.

Table 4 and Table 5, include some confusing values (i.e. .907, .906, ...etc). what does they mean? Please make sure to explain this issue in the manuscript.  

In Table 6, what are the applied Equations herein to conclude this sequence? The same is applicable in Table 7.

In the conclusion section, the author needs to highlight the novelty and aims of the study again, as well as the applied methods before presenting the output results, limitations, and recommendations for future works. Besides, the author has to present the effective contribution of this work to the thermal comfort and energy savings in construction projects. This aspect must be enhanced in conclusions.

Reviewer 3 Report

This is an interesting and detailed paper which provides insight into suitable indoor environments for the Thai elderly in hot season with an aim to develop guidelines for improved indoor environmental quality and reduced energy use. The topic is worthy of research in the light of rising ageing populations in Asia and the aspiration to reduce energy usage in the long term to promote sustainable development.

As to an overall evaluation, the study is acceptable after revisions, by considering the following suggestions:

While describing and setting up the 3 MSC’s the paper speaks about orientation, window sizes and potential solar gains from roofs. This research’s applicability would be further benefited if in the results and discussion section the authors were able to relate their findings back to these built environment parameters. The research could provide further recommendations, in line with building orientation, window sizes, etc. in the context of Thailand, so as to provide guidelines that promotes the use of environmental design strategies in the architectural design process while developing spaces for older people.

Reviewer 4 Report

The article aims to define the preferable comfort conditions for an elderly person, which is all very interesting considering the average age of the population is rising.

The article contains a number of generalisations because, in order to actually evaluate energy consumption, one should learn more about the built asset in terms of location, architecture, and form.  It is impossible to establish guidelines without this interdisciplinary knowledge, as is expressed beginning with the paper's title.

The globalisation of population and architecture taking place on a worldwide scale goes against sustainability with respect to all parameters.

Therefore, the authors are asked to specify the parameters to be adopted to achieve maximum comfort using the least amount of energy, based on the characteristics of the buildings, in further detail.

Round 2

Reviewer 2 Report

The author has answered most of the comments. A minor revision is required to answer the following:

In Lines 240 to 243, the author needs to identify the following terms: TSV, HSV, WSV, MTSV, MHSV, and MWSV. You need to clearly expalin what each concept means.

In lines 400 to 407, you have to cite references for the information related to sc STREAM, and DOE-2. 

Reviewer 4 Report

The changes and the new writing part introduced by the paper are good.
